# Gender Specificity of Spousal Concordance in the Development of Chronic Disease among Middle-Aged and Older Chinese Couples: A Prospective Dyadic Analysis

**DOI:** 10.3390/ijerph18062886

**Published:** 2021-03-11

**Authors:** Jing Liao, Jing Zhang, Jinzhao Xie, Jing Gu

**Affiliations:** 1Department of Medical Statistics, School of Public Health, Sun Yat-Sen University, Guangzhou 510080, China; liaojing5@mail.sysu.edu.cn (J.L.); zoejing27@outlook.com (J.Z.); xiejzh8@mail2.sysu.edu.cn (J.X.); 2Sun Yat-Sen Global Health Institute, School of Public Health, Sun Yat-Sen University, Guangzhou 510080, China

**Keywords:** gender specificity, couple, health concordance, chronic disease, dyadic data

## Abstract

This study aimed to explore the gender specificity of spousal concordance in the development of chronic diseases among middle-aged and older Chinese couples. Data of 3420 couples were obtained from the China Health and Retirement Longitudinal Study (CHARLS). Multivariate logistic regression was used to analyze the incidence of chronic disease development over 4 years, conditional on the spousal baseline chronic disease status; and stepwise adjusting for the couples’ sociodemographic characteristics (i.e., age, education, retirement status and household income), and their individual lifestyle (i.e., smoking, drinking, exercise, social participation and BMI) all measured at baseline. The incidence of chronic diseases after 4 years of follow-up was 22.95% in the husbands (605/2636) and 24.71% in the wives (623/2521). Taking the couples’ baseline sociodemographic and lifestyle covariates into account, husbands whose wife had chronic diseases at baseline showed an increased risk of developing chronic diseases over 4 years (OR_adjusted_ = 1.24, 95%CI: 1.02, 1.51), but this risk was not statistically-significant for wives (OR_adjusted_ = 0.88, 95%CI: 0.71, 1.08). Our study identified gender specificity of spousal concordance in the development of chronic diseases among middle-aged and older Chinese couples. This finding may contribute to the design of couple-based intervention for disease prevention and management for community-dwelling older adults.

## 1. Introduction

China has the world’s largest number of older adults, estimated at 248 million in 2020 and projected to rise to 482 million by 2053 [1,2]. In addition to the accelerated population ageing process, the burden of chronic diseases continues to grow. Chronic diseases are the leading causes of premature death [3] and impose a heavy burden on society and family. It is estimated that the number of middle-aged and older patients with chronic diseases will triple to 300 million by 2050 [4]. Furthermore, a 2016 nationwide survey of older Chinese adults recorded 4363 ‘empty-nest’ older couples (37.9% of the 11,511 older adults investigated) who relied mainly on each other for daily care [5].

Spousal concordance, that is, consistency in health status between a husband and a wife, has been well documented [6]. A range of cross-sectional studies on spousal concordance in chronic disease has been conducted worldwide, with evidence primarily of lifestyle-related chronic diseases, such as hypertension [7,8], cardiovascular disease [9,10,11], cancer [12,13] and type 2 diabetes [14,15]. This spousal concordance in chronic disease may be explained by assortative mating and the shared resources hypothesis. Assortative mating suggests that individuals tend to choose a spouse with a similar demographic profile, lifestyle, and life attitudes [16], whilst the shared resources hypothesis states that a couple’s shared living environment, economic resources [6,15], lifestyles [17,18,19] and social networks [20] may lead to their convergence in health status.

In line with the proposed hypotheses, cultural differences in spousal concordance in chronic disease have been identified [21]. Couples’ relationships may be influenced by different cultures; for example, an individual-oriented culture emphasizes independence and individual autonomy [22], whilst a collectivistic culture values connectedness and harmonious relationships [23]. Few Chinese studies have examined spousal concordance [17,24,25]. In line with the studies from other countries, these cross-sectional studies discovered spousal concordance in chronic disease. However, the study by Jurj et al. [17] was limited to women in Shanghai, and the information of their spouses was reported by the investigators instead of using paired data. Sun et al. [25] used community-based and national representative data, but the chronic disease was limited to diabetes. Both were cross-sectional studies that were unable to determine the chronological sequence of events and clear causal relationships. To sum up, Chinese studies focused only on cross-sectional study; rather, a cohort study with dyadic data would facilitate the understanding of concordance in the development of chronic disease among Chinese couples.

Moreover, gender differences in disease concordance within a couple have been suggested, given that the distinction of independency vs. interdependency is shaped not only by culture but also by gendered roles, but evidence regarding the gender specificity of spousal concordance remains limited and inconclusive. Some studies found no gender specificity [10,25,26], whilst others reported that husbands were more likely to be affected by their female spouses with regard to obesity, physical health and well-being [27,28,29]; in contrast, some showed that wives were more likely to be affected by their husband with regard to depression and chronic diseases [21,30,31,32]. To the best of our knowledge, the only prospective study in a Chinese cohort, which included 483 middle-aged and older adults from the Taiwan Longitudinal Study on Ageing (TLSA; year: 2000–2006), showed that the effects of spousal health status on the personal development of a cardiovascular disease or metabolic syndrome were only observed in wives [31]. Therefore, the cohort studies on the gender specificity of spousal concordance have not yielded a consistent conclusion. Paired data were rarely used, less focus was placed on the development of chronic diseases, and studies with larger samples were not conducted among middle-aged and older Chinese couples.

This prospective cohort study aimed to investigate the gender specificity of spousal concordance in the development of chronic diseases in middle-aged and older Chinese couples. We hypothesized that (1) the development of chronic diseases would co-occur among these couples, such that the participants whose spouse had a chronic disease at baseline would experience an increased risk of chronic disease development during the follow-up period relative to those with a healthy spouse; and (2) the extent to which husbands’ and wives’ development of chronic disease would be affected by their sick partners would differ.

## 2. Materials and Methods

### 2.1. Study Samples

Our study sample was derived from the China Health and Retirement Longitudinal Study (CHARLS 2011–2015), a nationally representative longitudinal study of community-dwelling older adults aged 45 years and above, with a follow-up interval of 2 years (Wave 1 [baseline, 2010–2011], Wave 2 [2012–2013] and Wave 3 [2014–2015]) [33]. The respondents were interviewed face-to-face in their homes via computer-assisted personal interviewing (CAPI) technology. Our sample comprised 3420 CHARLS couples that fulfilled the following criteria: (1) both spouses were included in the data; (2) both spouses were at least 45 years old at baseline; (3) both had complete data on chronic disease status at baseline (Wave 1, 2010–2011) and at the end of the 4-year follow-up (Wave 3, 2014–2015). Because our study’s aim was to explore spousal mutual influences on chronic disease development, couples in which both spouses had chronic diseases at baseline were excluded.

### 2.2. Measurement

#### 2.2.1. Chronic Disease Status

The respondents were asked whether they had received a diagnosis of any of the following chronic diseases in Wave 1 (2010–2011) and Wave 3 (2014–2015): hypertension, diabetes or high blood sugar, cancer and malignant tumor (excluding small skin cancer), chronic lung disease (e.g., emphysema and chronic bronchitis), cardiac disease (e.g., myocardial infarction, coronary heart disease, angina pectoris, heart failure and other heart problems), stroke, and dyslipidemia (elevation of low-density lipoprotein-cholesterol, triglycerides (TGs), and total cholesterol, or reduction of high-density lipoprotein-cholesterol) [33]. Accordingly, chronic disease status was assigned according to the answers to these questions, with ‘0’ referring to the absence of chronic disease and ‘1’ referring to the presence of at least one of these seven chronic diseases. All these chronic diseases were based on self-reports.

#### 2.2.2. Chronic Disease Development

A binary variable was created to represent chronic disease development over 4 years (from Wave 1 to Wave 3). A change in the chronic disease status (‘1’) meant a change from no chronic disease to at least one studied chronic disease, whilst unchanged chronic disease status (‘0’) included the absence or presence of chronic disease at both waves.

#### 2.2.3. Covariates

Sociodemographic covariates at baseline (Wave 1) included age (in years; centered at 56.55 years), gender (men = 0; women = 1), education (lower secondary education = 0; upper secondary education = 1), retirement status (in-service = 0; retired = 1) and household total income (in thousand RMB, the sum of all annual income at the couple level).

#### 2.2.4. Lifestyle

Key lifestyles were selected according to the spousal health concordance literature [6,15,18,20] and included smoking, drinking, exercise, social participation and (pre-)obesity at baseline (Wave 1). Smoking was identified via a ‘yes’ response to the question ‘still chewed tobacco, smoked a pipe, smoked self-rolled cigarettes or smoked cigarettes/cigars?’ (no = 0; yes = 1). Drinking was identified via a ‘yes’ response to the question ‘Did you drink any alcoholic beverages, such as beer, wine or liquor in the past year?’ (no = 0; yes = 1). Exercise [34,35] was identified via a ‘yes’ response to the question ‘During a usual week, did you do any moderate or vigorous activities for at least 10 min continuously? (moderate physical activities [e.g., bicycling and carrying light loads] or vigorous physical activities [e.g., heavy lifting, digging, ploughing, aerobics and fast bicycling]) (no = 0; yes = 1).’ Social participation was identified via a ‘yes’ response to the question ‘Have you done any of these activities in the last month? Interacted with friends; played mahjong, chess or cards; or went to community club and so on’ (no = 0; yes = 1). Furthermore, underweight and (pre-)obesity [36] was classified by BMI (body mass index, kg/m^2^) (normal weight:18.5–24 = 0; underweight: less than 18.5 = 1; (pre-)obesity: more than 24 = 2), which to some extent reflected diet behaviors [37]. The participants’ BMI was calculated as the mean of three measures of body weight in kilograms divided by the mean of height in meters squared (kg/m^2^) by a trained staff member using a health meter (Omron HN-286, Omron Corp., Kyoto, Japan) and a stadiometer (Seca^TM^ 213, SECA Corp., Hamburg, Germany).

## 3. Statistical Analysis

We used McNemar’s chi-square test and paired-sample t-test to examine the differences between spouses and a Kappa test to examine the associations between husbands and wives. The strength of the Kappa coefficients is interpreted in the following manner: 0.01 to 0.20 is slight; 0.21 to 0.40 is fair; 0.41 to 0.60 is moderate; 0.61 to 0.80 is substantial; and 0.81 to 1.00 is almost perfect. Pearson’s chi-square test (for two independent samples) was used to test the difference in the development of chronic disease between husbands and wives and to test the differences in the development of chronic disease between spouses with and without chronic diseases at baseline among husbands and wives.

The association between spousal chronic disease status at baseline and personal development of chronic disease over 4 years was estimated with logistic regression models for husbands and wives separately. In addition, participants who had a chronic disease at baseline were excluded from the logistic regression analysis, whereby 2521 couples and 2636 couples were included in the logistic regression among wives and husbands, respectively (Figure 1). Four adjustment models were developed: Model 0 was unadjusted; Model 1 adjusted the ages of both spouses; Model 2 further adjusted other sociodemographic covariates (education level, retirement status and household income) of both spouses at baseline; in Model 3, personal lifestyles (smoking, drinking, social participation, exercise and BMI]) at baseline were adjusted; and in Model 4, we further adjusted for spousal lifestyle (smoking, drinking, social participation, exercise and BMI at baseline).

We used Stata MP 14.0 (College Station, TX, USA, 2014) to perform all analyses, with the significance level set as α = 0.05.

## 4. Results

### 4.1. Baseline Characteristics of Participants

A total of 5793 couples met the three inclusion criteria and were selected as the subjects of this study, but 2373 couples were excluded because both spouses had received a diagnosis of a chronic disease at baseline. Finally, 3420 couples were included in the analysis. Among the 3420 couples at baseline (Figure 1), both spouses were free of chronic disease in 50.79% couples, only the husband had a diagnosed chronic disease in 22.92% couples, and only the wife had a diagnosed chronic disease in 26.29% couples.

As shown in Table 1, among the 3420 couples, the husbands were older, better educated and less likely to be retired than their wives (all *p* < 0.001). Baseline lifestyles also showed that the husbands were more likely to smoke, drink alcohol, exercise and underweight, but they were less likely to have (pre-) obesity than their wives (all *p* < 0.001). The Kappa coefficient showed only fair agreement within couples on social participation (Kappa coefficient = 0.36; *p* < 0.001), but slight agreement on smoking, drinking, exercise and (pre-) obesity. At baseline, 22.92% of husbands and 26.29% of wives reported at least one chronic disease.

### 4.2. Chronic Disease Development

After 4 years of follow-up, among those who had no chronic disease at baseline, the incidence of chronic disease was 22.95% in husbands (605/2636) and 24.71% in wives (623/2521; *p* = 0.002) (Table 1). Of these, the list of specific chronic disease among those who newly developed chronic diseases after 4 years were shown in Table 2.

### 4.3. Gender Specificity of Spousal Concordance in Chronic Disease Development

The husbands whose wife had a chronic disease at baseline reported a higher incidence of chronic disease at Wave 3 than those whose wife did not have a chronic disease at baseline (21.47% vs. 25.81%; *p* = 0.012, data not tabulated). While no difference was shown in women (24.87% vs. 24.36%; *p* = 0.784).

In the multivariate analyses for husbands (Table 3), after adjustment for different sets of covariates, the association between spousal (wives’) baseline chronic disease status and the husbands’ development of chronic disease remained significant (OR_adjusted_ range, 1.23 to 1.27; *p* < 0.05).

In similar analyses for wives (Table 3), after adjustment for different sets of covariates, no significant association was discovered between spousal (husbands’) baseline chronic disease status and the wives’ development of chronic disease (OR_adjusted_ range, 0.88 to 0.97; *p* > 0.05).

## 5. Discussion

Based on 3420 pairs of middle-aged and older Chinese couples of a national representative ageing cohort, we found that almost one fourth of study participants developed at least one chronic disease over 4 years of follow-up, and hypertension was the most common newly developed chronic diseases. Moreover. After adjustment for baseline confounders, husbands whose wives had a chronic disease at baseline had an increased risk of chronic disease development over time, but this risk was not significant for wives conversely.

Our study contributes to the literature of spousal health concordance by investigating the development of chronic disease over four years conditional on spousal chronic diseases, stratified by gender and adjusted for covariates of both couples. Few studies previously had investigated spousal concordance in chronic disease development, which mainly focused on concurrent association of health status [6,10,38] and health behaviors [6,39,40]. Our finding of the spouse-dependent onset of chronic diseases was consistent with recent studies about the spousal concordance in obesity development [28], changes in biomarker and subjective well-being [31], and mental health and self-reported health trajectories [29]. We extended the evidence in a Chinese community-based setting, with lifestyle data directly collected from both couples, and health status defined by multiple chronic disease status that were more common in older adults [41].

In view of the inconsistent findings on gender variation [29,30], we further advanced the literature by investigating spousal health concordance gender-specifically. In general, previous studies of married couples have shown that the health benefits of marriage and the health detriments of marital dissolution differ by gender [6,42,43]. Our study showed that the influence of the spouse’s health on the other was only evident in husbands but not wives. Cultural differences maybe the important reason for the gender-specific association of chronic disease development [21]. Our finding was different from some studies based in the UK and The Netherlands, showing that wives were more likely to be impacted by their spouses’ chronic status psychologically and physically than husbands [30,32]. On the other hand, other UK-based studies did reveal similar findings as ours that husbands but not wives experienced declines in self-reported health after their spouse’s onset of chronic diseases [29], and husbands’ but not wives’ obesity development was associated with their spouses’ diabetes status [28]. Conflicting results may indicate other subtle influences between and characteristics of the couples beyond the general stereotype of culture.

Other possible reasons for our gender-specific finding may result from as follows. Wives play a predominant role of caregiver in the family under greater socialization factors [44], which is practically true under the traditional Chinese gendered social role, namely ‘breadwinning men and homemaking women’ [41,45]. Husbands may be more dependent on their spouses regarding lifestyles and health management, such that husbands whose spouse with chronic diseases may suffer more chronic disease risk than their female counterparts. Moreover, as suggested by literature men tend to maintain intimate relationships with fewer people and receive social support primarily from their spouses [46,47], while women are more likely to have broader social support other than that of their spouses [46,48]. Retired Chinese women are more socially-engaged than their male counterparts, actively participating in social- and physical- group activities, such as square-dancing [49]. While wives’ lifestyle can be also influenced by their peers [49], their husbands’ health status may be mostly influenced by their wives’ lifestyle and lifestyle-related chronic disease status [50,51].

These gender-specific association of chronic disease development was largely independent of sociodemographic and lifestyle factors included. Possible reasons for these non-significant adjustments may result from limited variations in our participants’ retirement status, education and exercise levels that may be underpowered to detect a statistically significant adjustment for the given sample size. Moreover, variations in lifestyles were further reduced as we only included baseline measures, which were likely to change as participants aged and be influenced by the older couples’ health status [52].

Given the greater frequency of chronic diseases and increased dependency that occur in later life [53], our finding shows that for chronic disease prevention in China, spousal concordance in the development of chronic disease could inform prevention advice that shifts the focus from optimizing prevention efforts for the individual patient alone to optimizing couple-based interventions. Moreover, spousal concordance could also be used for earlier detection of chronic diseases to make people recognize and respond to health problems earlier and be more willing to undergo treatment [15,54]. Our finding of gender specificity also indicates that the diagnosis of a chronic disease in one spouse may warrant increased surveillance in his/her partner, because a husband whose wife has a chronic disease may obtain benefit from such increased surveillance.

With a prospective dyadic design stratified by gender, our study explored gender specificity in the spousal concordance of chronic disease development over 4 years based on a population-representative sample of middle-aged and older Chinese couples. Several limitations warrant notice. First, our study used baseline lifestyles to obtain their clear temporal relationship with the incidence of chronic disease during the follow-up period, which overlooked lifestyle changes over time. Further studies are needed to better understand the dynamics between lifestyle and chronic disease development with age. Second, we defined chronic disease status in a purely qualitative manner (i.e., at least one of seven chronic diseases) rather than by quantity or specified disease. Metabolic and cardiovascular diseases may be more susceptible to influence from lifestyle than other chronic diseases, and multiple chronic diseases and single chronic diseases should be discussed separately. More other diseases (i.e., alcoholic liver disease) were unavailable. Thus, further investigation of more comprehensive, specified disease type and quantity is needed. In addition, our study primarily used self-reported outcome and lacked other related lifestyles (i.e., sleeping time, diet habit and relaxing time together). To reduce information bias and further explore the mechanism, more lifestyles, objective measures such as biomarkers, and physician verification of chronic disease status are needed to improve the accuracy of the findings.

## 6. Conclusions

This study revealed gender-specific patterns of spousal concordance in the development of chronic disease among community-dwelling middle-aged and older Chinese couples and demonstrated that husbands whose wife had a chronic disease at baseline had an increased risk of developing a chronic disease over the next 4 years, but this risk was not significant for wives. Our finding may inform the design of couple-based chronic disease management for older adults to better serve the health needs of the ageing population, whereby the health status and risk factors of both spouses would be jointly examined. Our findings suggest that women should be supported and men motivated for chronic disease surveillance if their spouse has a chronic disease.

## Figures and Tables

**Figure 1 ijerph-18-02886-f001:**
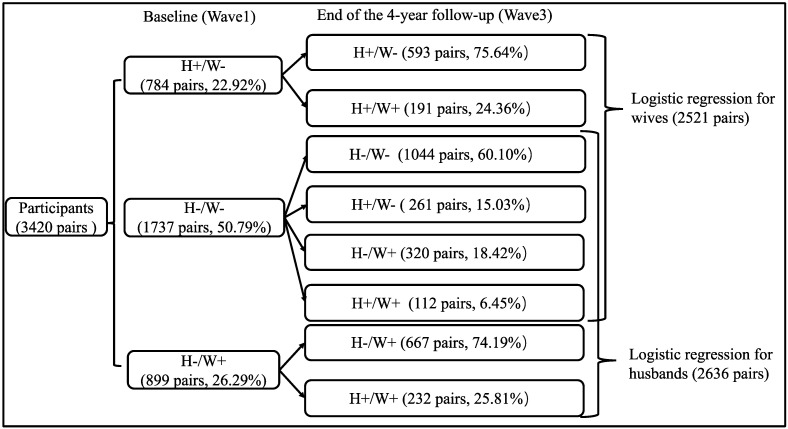
Flowchart of chronic disease development from baseline to 4-year follow-up among couples. Notes. H: husbands; W: wives; -: no chronic diseases; +: with chronic diseases.

**Table 1 ijerph-18-02886-t001:** Baseline characteristics and chronic disease status at Wave 1 and Wave 3 by gender (3420 couples).

	Husbands	Wives	*p* (for Difference)	Kappa	*p* (for Kappa)
	*n* = 3420	*n* = 3420			
	%	%			
**Sociodemographic characteristics at baseline**					
Age, year [Mean (SD)]	57.54 (8.54)	55.61 (8.01)	<0.001	0.12	<0.001
Education level, Upper secondary	17.31	8.89	<0.001	0.28	<0.001
Retired, yes	16.20	24.50	<0.001	0.39	<0.001
Household income, thousand [Mean (SD)]	33.57 (66.99)		-	-	-
**Lifestyles at baseline**					
Smoking, yes	60.29	5.67	<0.001	0.01	0.047
Drinking, yes	59.94	11.81	<0.001	0.06	<0.001
Social participation, yes	48.07	46.14	0.05	0.36	<0.001
Exercise, yes	77.08	73.98	<0.001	0.19	<0.001
Underweight (BMI < 18.5 kg/m^2^), yes	5.88	5.56	<0.001	0.05	<0.001
(Pre-)obesity (BMI ≥ 24 kg/m^2^), yes	32.84	44.24	<0.001	0.05	<0.001
**Chronic disease status ^2^**					
Wave 1 (baseline, 2010–2011), yes	22.92	26.29	0.001	-	-
Wave 3 (2014–2015), yes	38.10	42.49	<0.001	-	-
**Chronic disease development (Wave 1 to 3) ^3^**					
Changed chronic disease condition, yes	22.95	24.71	0.002 ^1^	-	-

^1^*p* value of chi-square test of two independent groups. ^2^ Chronic disease status: ‘yes’ refers to the presence of at least. one of seven chronic diseases: high blood pressure, diabetes or high blood sugar, cancer, chronic lung diseases, heart disease, stroke, and dyslipidemia. ^3^ Chronic disease development: Among those who had a negative chronic disease status at baseline (2636 husbands, 2521wives). ‘yes’ indicates that the chronic disease status had changed from disease-free to the presence of at least one of nine chronic diseases over 4 years (from Wave 1 to Wave 3). Abbreviations: SD, Standard Deviation.

**Table 2 ijerph-18-02886-t002:** The list of specified chronic disease among newly developed chronic diseases over 4 years.

Chronic Disease	N (Person)	%
**For husbands (*n* = 605)**		
Hypertension	276	45.62
Dyslipidaemia	172	28.43
Lung disease	136	22.48
Cardiac disease	101	16.69
Diabetes	63	10.41
Stroke	28	4.63
Cancer	9	1.49
**For wives (*n* = 623)**		
Hypertension	271	43.50
Dyslipidaemia	185	29.70
Lung disease	148	23.76
Cardiac disease	94	15.09
Diabetes	80	12.84
Stroke	17	2.73
Cancer	17	2.73

**Table 3 ijerph-18-02886-t003:** Influence of spousal baseline chronic disease status on one’s chronic disease development over 4 years.

Spousal Baseline Chronic Disease Status	Chronic Disease Development over 4 Years (Yes)					
	%	Model 0 ^1^	Model 1 ^2^	Model 2 ^3^	Model 3 ^4^	Model 4 ^5^
		OR (95% CI)	OR_a_ (95% CI)	OR_a_ (95% CI)	OR_a_ (95% CI)	OR_a_ (95% CI)
**For husbands (*n* = 2636)**						
No	21.47	1.00	1.00	1.00	1.00	1.00
Yes	25.81	1.27 (1.05,1.54) **	1.24 (1.02,1.50) *	1.23 (1.02,1.49) *	1.25 (1.03,1.51) *	1.24 (1.02,1.51) *
**For wives (*n* = 2521)**						
No	24.87	1.00	1.00	1.00	1.00	1.00
Yes	24.36	0.97 (0.80,1.18)	0.92 (0.75,1.12)	0.92 (0.75,1.12)	0.90 (0.73,1.10)	0.88 (0.71,1.08)

^1^ Model 0: unadjusted model. ^2^ Model 1: adjusted for age of both spouses. ^3^ Model 2: adjusted for age of both spouses and other sociodemographic covariates (education level, retirement status and household income) of both spouses. ^4^ Model 3: adjusted for sociodemographic covariates (age, education level, retirement status and household income) and both spouse’s personal lifestyles (smoking, drinking, social participation, exercise and BMI). ^5^ Model 4: adjusted for covariates in Model 3 (sociodemographic covariates of both spouse’s and personal lifestyles) and spousal lifestyles (smoking, drinking, social participation, exercise and BMI). OR_a_: OR adjusted. **: *p* < 0.01; *: *p* < 0.05.

## Data Availability

The datasets analyzed in this study are available from The China Health and Retirement Longitudinal Study (CHARLS 2011–2015) repository, and the searchable website www.g2aging.org (accessed on 5 August 2020) can be used to access the CHARLS 2011–2015 data.

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
