# Peer review of "Gender Specificity of Spousal Concordance in the Development of Chronic Disease among Middle-Aged and Older Chinese Couples: A Prospective Dyadic Analysis"

_ijerph, 2021, doi:10.3390/ijerph18062886_

Round 1

Reviewer 1 Report

This manuscript examining spousal concordance in chronic disease development over time was a pleasure to read. The writing is clear and concise, the methods appropriate, the discussion comprehensive and interesting, and the conclusions sound. I have no suggestions for improving this manuscript.

(One typo at line 243: "paly" should be "play")

Author Response

Reviewer reports:

Reviewer 1: Reviewer's comments

  1. One typo at line 243: "paly" should be "play"

Responses: Thank you very much for your confirmation of our study analysis. And thank you very much for pointing this typo out. We have revised the text accordingly. (Line 262, page 7)

Reviewer 2 Report

See document attached below.

Author Response

Reviewer reports:

Reviewer 2: Reviewer's comments

Major 1. The authors defined chronic diseases as the presence of at least one of 9 disease: high blood pressure, DM, cancer, lung disease, cardiac disease, stroke, emotional or psychiatric problems, arthritis and dyslipidemia. Why did the authors included “emotional or psychiatric problems” and “arthritis” in this study. They are not lifestyle-related disease and seems different from other chronic diseases. Describe the rationale for inclusion or remove the two diseases from analysis.

Responses: Thank you very much for your valuable comment. We take your suggestion that the inclusion of emotional or psychiatric problems, and arthritis was inappropriate in this study, and rerun our analysis accordingly. The results of the analysis didn’t change significantly from the previous one. We changed the corresponding texts as, for instance: in the Abstract,” The incidence of chronic diseases after 4 years of follow-up was 22.95% in the husbands (605/2636) and 24.71% in the wives (623/2521). Taking the couples’ baseline sociodemographic and lifestyle covariates into account, husbands whose wife had a chronic disease at baseline showed an increased risk of developing a chronic disease over 4 years (ORadjusted=1.24, 95%CI:1.02,1.51), but this risk was not statistically-significant for wives (ORadjusted=0.88, 95%CI:0.71,1.08). (Line 20-25, page1)

The Methods (Line 111-118, page3; Line 161-164, page4), Figure 1(Line 172, page4), Table 1(Line 191, page5), Table 3 (Line 220, page6) and Results (Line 181-182; Line 189; Line 199-201; Line 206-207; Line 210-211; Line 214-215Line 229., page 5-7) have also been revised.

Major 2. Add Table 3, showing the number of participants who have newly developed chronic diseases and its breakdown to at least top 5, like below. This table will be helpful on implementation of the couple-based intervention.

Responses: Thank you very much for your valuable comment. We have added this Table 2Table 2,Line 217, page6and the brief description in Results and Discussion accordingly as below.

Table 2. The list of specified chronic disease among newly developed chronic diseases over 4 years

Chronic disease

Nperson

%

For husbands (n=605)

Hypertension

276

45.62

Dyslipidaemia

172

28.43

Lung disease

136

22.48

Cardiac disease

101

16.69

Diabetes

63

10.41

Stroke

28

4.63

Cancer

9

1.49

For wives (n=623)

Hypertension

271

43.50

Dyslipidaemia

185

29.70

Lung disease

148

23.76

Cardiac disease

94

15.09

Diabetes

80

12.84

Stroke

17

2.73

Cancer

17

2.73

Of these, the list of specific chronic disease among those who newly developed chronic diseases after 4 years were shown in Table 2.” (Line 201-202, page5)

and hypertension was the most common newly developed chronic diseases.” (Line 230-231, page6)

Major 3. Did authors write about the ethical statement?

Responses: Thank you very much for pointing this out. We have added an Ethical Considerations in the revised manuscript as below.

“The original CHARLS study was approved by the Biomedical Ethics Review Committee of Peking University (IRB00001052–11015), and all interviewees were required to sign informed consent at the time of participation. Ethics approval for the use of CHARLS data was obtained from the University of Newcastle Human Research Ethics Committee (H-2015-0290).” (Line 102-107, page 3)

Minor 1. Line 85, is aim 2) necessary? Aim 1) seems enough.

Response: Thank you very much for pointing out this confusing point. But we list these two aims separately following considerations. First, the first aim was to present a broad phenomenon of spousal concordance in chronic disease, i.e., whether one's own chronic disease development was affected by one's spouse's chronic disease status. While the second aim was the further inquiry into whether this effect were gender differences. In addition, since we mentioned in the introduction that the cohort studies on the gender specificity of spousal concordance have not yielded a consistent conclusion [1-3]. Therefore, both in the introduction, results and discussion of the manuscript, we had presented and compared them with different genders, so as to highlight the second aim.

Therefore, we kept two aims to ensure the complete framework of the article.

Minor 2. Line 102, was this study conducted by interview or questionnaire? Add explanation.

Responses: Thank you very much for pointing this out. We have added the explanation in the revised manuscript as below.

The respondents were interviewed face-to-face in their homes via computer-assisted personal interviewing (CAPI) technology.” (Line 93-95, page 2)

Minor 3. Wasn’t household income “equivalised income”? e.g., total income / square root (family number).

Response: Thank you very much for pointing out this confusing point. In CHARLS study, the household total income was the sum of all income at the couple level including income of both spouses from earning income, capital income, pension income, income from government transfers and other income. It wasn’t the equivalized income. In previous publications using CHARLS data, the household total income was also used to measure the income level of couples or individuals [4]. We have added the brief explanation of household total income in the revised manuscript as below.

household total income (in thousand RMB, the sum of all annual income at the couple level)” (Line 128-129, page 3)

Minor 4. Line129, why was “10 minutes” cut-off point in exercise? If some reasons exist, add it.

Response: Thank you very much for pointing out this confusing point. First, the CHARLS had used the Global Physical Activity Questionnaire (GPAQ), a standardized tool to measure physical activity provided by WHO [5], and around 50 developing countries are now using GPAQ for physical activity data collection. In addition, The World Health Organization (WHO) guidelines stated that in order to stay healthy and improve health, adults aged 18–64 years should perform at least 150-minute of moderate-intensity aerobic physical activity or at least 75-minute of vigorous-intensity aerobic physical activity throughout the week, with each aerobic activity performed in bouts of at least 10-minute duration [6]. Thus, the 10-minute criterion was also chosen in our study. We have added the references of exercise measurement in the revised manuscript. (Line 137, page 3)

Minor 5. Line 170, on participants, add the follow-up rate.

Responses: Thank you very much for pointing this out. First, our study was analysed based on the CHARLS study, with a response rate of 80.5% and a follow-up rate of 91.0 % in the original CHARLS study [7], which was among the higher of similar international survey programs. In addition, the inclusion criteria of our study samples specified that “3) both had complete data on chronic disease status at baseline (Wave 1, 2010–2011) and at the end of the 4-year follow-up (Wave 3, 2014–2015)”. (Line 97-98, page 3), so the follow-up rate of the participants in our analysis was 100% and would not have been listed in the manuscript.

Minor 6. Line 196, important data should be included in table 2, like below (in the attachment).

Responses: Thank you very much for your valuable comment. we have included this important information in Table 3 (original Table 2) of the revised manuscript. (Table 3, Line 220, page 6)

Minor 7. Line 243, typo in “paly”.

Responses: Thank you very much for pointing this typo out. We have revised the text accordingly. (Line 262, page 7)

Minor 8. Line 251, Check “Chinese retired women”. I think “retired Chinese women” may be correct.

Responses: Thank you very much for pointing this out. We have revised the text accordingly. (Line 270, page 7)

References

[3] Ramezankhani A, Guity K, Azizi F, Hadaegh F. Sex differences in the association between spousal metabolic risk factors with incidence of type 2 diabetes: a longitudinal study of the Iranian population. Biology of sex differences. 2019;10(1):41.

[4] Ke C, Qiao Y, Liu S, Rui Y, Wu Y. Longitudinal research on the bidirectional association between depression and arthritis [published online ahead of print, 2020 Nov 22]. Soc Psychiatry Psychiatr Epidemiol. 2020. doi:10.1007/s00127-020-01994-7

[5] Armstrong T, Bull F. Development of the World Health Organization Global Physical Activity Questionnaire (GPAQ). J Public Health. 2006;14: 66–70 .

[6] World Health Organization. Global physical activity questionnaire (GPAQ) analysis guide. Geneva: World Health Organization. 2012.

[7] Chen X, Smith J, Strauss J, Wang Y, Zhao Y. China Health and Retirement Longitudinal Study (CHARLS). Encyclopedia of Geropsychology. 2015. https://doi.org/10.1007/978-981-287-080-3_42-1

Reviewer 3 Report

The study has treated with an interesting topic, while there were several points to be considered.

  1. The findings may differ among countries and ethnicities. This can be more discussed from multi-aspects with more citations (culture…).
  2. In methodology, why did the authors use OR, not hazard ratios, in the cohort study? The theory can be described in Methods.
  3. In Table, the unadjusted OR (hazard ratios) should be first demonstrated.
  4. In Methods, a lean BMI should be considered for the development of chronic diseases in older people. The categories can be added.
  5. In Methods, the definition of chronic disease status should be more detailed. More other diseases such as alcoholic liver disease can be included.
  6. The types of arthritis should be defined.
  7. The types of dyslipidemias should be defined.
  8. The types of emotional and psychiatric diseases should be defined.
  9. The mechanisms were poorly understood. Did you have the more data of linked lifestyles, e.g., sleep time in each other, relaxing time together at home…?
  10. It is better to receive for native check again.

Author Response

Reviewer reports:

Reviewer 3: Reviewer's comments

  1. The findings may differ among countries and ethnicities. This can be more discussed from multi-aspects with more citations (culture…).

Responses: Thank you very much for your valuable comment. We agree with your that we should discussed more among different countries and ethnicities. We have added more discussion about gender-specific association of chronic disease development in the revised manuscript as below.

“Cultural maybe the important reason for the gender-specific association of chronic disease development [1]. Our finding was different from some studies based in the UK and the Netherlands, showing that wives were more likely to be impacted by their spouses’ chronic status psychologically and physically than husbands [2-3]. On the other hand, other UK-based studies did reveal similar findings as ours that husbands but not wives experienced declines in self-reported health after their spouse’s onset of chronic diseases [4], and husbands’ but not wives’ obesity development was associated with their spouses’ diabetes status [5]. Conflicting results may indicate other subtle influences between and characteristics of the couples beyond the general stereotype of culture.” (Line 251-260, page 7)

  1. In methodology, why did the authors use OR, not hazard ratios, in the cohort study? The theory can be described in Methods.

Responses: Thank you very much for pointing out this confusing point. We agree with you that the Cox model (hazard ratios, HR) is superior to the logistic model for cohort study by taking into account exact individual times of failure and censored cases [6]. However, the participants included in our study had almost the same follow-up time of 4 years, and had not censored. Therefore, we chose the more suitable method, Logistic model (odds ratios, OR) to analysis for our study.

  1. In Table, the unadjusted OR (hazard ratios) should be first demonstrated.

Responses: Thank you very much for your valuable comment. we have added the unadjusted OR in the Table 3 of the revised manuscript. (Table 3, Line 220, page 6)

  1. In Methods, a lean BMI should be considered for the development of chronic diseases in older people. The categories can be added.

Responses: Thank you very much for your valuable comment. We have added the category of underweight and classified BMI into 3 categories based on Chinese criteria [7]. Method and Result of the manuscript have been revised accordingly as below.

“Furthermore, underweight and (pre-)obesity [7-8] was classified by BMI (body mass index, kg/m2) (normal weight:18.5-24=0; underweight: less than 18.5 =1; (pre-)obesity: more than 24=2), which to some extent reflected diet behaviors [9].” (Line 143-146, page 3-4)

“Baseline lifestyles also showed that the husbands were more likely to smoke, drink alcohol, exercise and underweight, but they were less likely to have (pre-) obesity than their wives (all P<0.001).” (Line 184-186, page 5)

  1. In Methods, the definition of chronic disease status should be more detailed. More other diseases such as alcoholic liver disease can be included.

Responses: Thank you very much for your valuable comment. We strongly agree with your point. However, our study was based on CHARLS, a nationally representative longitudinal survey of the middle-aged and elderly population of China, including assessments of the social, economic, and health circumstances of community residents [10]. In addition, chronic diseases were based on self-reports and the crude disease categories since the original study was too extensive. Thus, many chronic diseases were not specified or not included (alcoholic liver disease was unavailable), which was the limitation of our study. We have added more detailed definition of chronic disease status and the limitations in the revised manuscript as below.

“Accordingly, chronic disease status was assigned according to the answers to these questions, with ‘0’ referring to the absence of chronic disease and ‘1’ referring to the presence of at least one of these seven chronic diseases. All these chronic diseases were based on self-reports.” (Line 116-119, page 3)

More other diseases (i.e., alcoholic liver disease) were unavailable. Thus, further investigation of more comprehensive, specified disease type and quantity is needed.” (Line 303-304, page 8)

  1. The types of arthritis should be defined.

Responses: Thank you very much for pointing this out. First, as we responded in comment 5, chronic diseases were based on self-reports and the crude categories since the original study was too extensive. So, the type of arthritis and emotional and psychiatric diseases were not defined clearly in the original CHARLS study. In addition, based on another reviewer's suggestion that remove arthritis, and emotional and psychiatric diseases from analysis, which are not lifestyle-related disease and seems different from other chronic diseases. Thus, we rerun our analysis accordingly. The results of the analysis didn’t change significantly from the previous one. We changed the corresponding texts as, for instance: in the Abstract,” The incidence of chronic diseases after 4 years of follow-up was 22.95% in the husbands (605/2636) and 24.71% in the wives (623/2521). Taking the couples’ baseline sociodemographic and lifestyle covariates into account, husbands whose wife had a chronic disease at baseline showed an increased risk of developing a chronic disease over 4 years (ORadjusted=1.24, 95%CI:1.02,1.51), but this risk was not statistically-significant for wives (ORadjusted=0.88, 95%CI:0.71,1.08). (Line 20-25, page 1)

The Methods (Line 111-118, page3; Line 161-164, page4), Figure 1(Line 172, page4), Table 1(Line 191, page5), Table 3 (Line 220, page6) and Results (Line 181-182; Line189; Line 199-201; Line 206-207; Line 210-211; Line 214-215Line 229., page5-7) have also been revised.

  1. The types of dyslipidemias should be defined.

Responses: Thank you very much for pointing this out. we have added the detailed definition of dyslipidemia in the revised manuscript as below.

“dyslipidemia (elevation of low-density lipoprotein, triglycerides (TGs), and total cholesterol, or a low high density lipoprotein level)” (Line 115-116, page 3)

  1. The types of emotional and psychiatric diseases should be defined.

Responses: Thank you very much for pointing this out. We have already responded in comment 6.

  1. The mechanisms were poorly understood. Did you have the more data of linked lifestyles, e.g., sleep time in each other, relaxing time together at home…?

Responses: Thank you very much for your valuable comment. We strongly agree with your point. First, as we responded in comment 5, our study was based on CHARLS, including assessments of the social, economic, and health circumstances of community residents, but the lifestyles related surveys were limited. We had included all lifestyle-related variables in our analysis, the other lifestyles like sleeping time, diet habit, relaxing time together at home were unavailable. We used to list it in the limitation and have further modified the limitation in the revised manuscript as below.

“In addition, our study primarily used self-reported outcome and lacked other related lifestyles (i.e., sleeping time, diet habit and relaxing time together). To reduce information bias and further explore the mechanism, more lifestyles, objective measures such as biomarkers, and physician verification of chronic disease status are needed to improve the accuracy of the findings.” (Line 304-309, page 8)

  1. It is better to receive for native check again.

Responses: Thank you very much for your valuable comments. We have revised the manuscript again for native check, and hope that the revised manuscript will satisfy you.

References

[1] Ramezankhani A, Guity K, Azizi F, Hadaegh F. Sex differences in the association between spousal metabolic risk factors with incidence of type 2 diabetes: a longitudinal study of the Iranian population. Biology of sex differences. 2019;10(1):41.

[2] Ayotte BJ, Yang FM, Jones RN. Physical Health and Depression: A Dyadic Study of Chronic Health Conditions and De-pressive Symptomatology in Older Adult Couples. Journals of Gerontology 2010; 65B: 438-448.

[3] Hagedoorn M, Sanderman R, Ranchor AV, Brilman EI, Ormel J. Chronic disease in elderly couples: are women more responsive to their spouses' health condition than men? Journal of Psychosomatic Research. 2001; 51: 693-696.

[4] Valle G, Weeks JA, Taylor MG, Eberstein IW. Mental and physical health consequences of spousal health shocks among older adults. Journal of aging and health. 2013;25:1121-1142.

[5] Silverman O, Hulman A, Simmons RK, Nielsen J, Witte DR. Trajectories of obesity by spousal diabetes status in the English Longitudinal Study of Ageing. Diabetic Medicine. 2019;36:105.

[6] Cree MS, Symons MJ. A comparison of the logistic risk function and the proportional hazards model in prospective epidemiological studies. J Chron Dis 1983,36:715-724.

[7] McNatty KP, Hudson NL, Collins F, Fisher M, Health DA, Henderson KM. Effects of oestradiol-17 beta, progesterone or bovine follicular fluid on the plasma concentrations of FSH and LH in ovariectomized Booroola ewes which were homozygous carriers or non-carriers of a fecundity gene. J Soc Reprod Fertil. 1989; 87:573–585.

[8] World Health Organization. Global physical activity questionnaire (GPAQ) analysis guide. Geneva: World Health Organization. 2012.

[9] Heerman WJ, Jackson N, Hargreaves M, Mulvaney SA, Schlundt D, Wallston KA, Rothman RL. Clusters of Healthy and Unhealthy Eating Behaviors Are Associated with Body Mass Index Among Adults. Journal of nutrition education and behavior. 2017; 49: 415-421.e411.

[10] Yaohui Zhao, Yisong Hu, James P Smith, John Strauss, Gonghuan Yang, Cohort Profile: The China Health and Retirement Longitudinal Study (CHARLS), International Journal of Epidemiology. 2014;43(1):61-68.

Round 2

Reviewer 2 Report

The authors revised the manuscript accordingly.

Author Response

Thank you very much for your confirmation of our study and for providing us with very useful and valuable comments.

Reviewer 3 Report

The paper has been much improved. Regarding dyslipidemias, the expression of low-density lipoprotein or high density lipoprotein should be in a preciously academic form. Not low-density lipoprotein but low-density lipoprotein-cholesterol should be used. High density should be changed to high-density.

Author Response

Reviewer reports:

Reviewer 3: Reviewer's comments

  1. The paper has been much improved. Regarding dyslipidemias, the expression of low-density lipoprotein or high density lipoprotein should be in a preciously academic form. Not low-density lipoprotein but low-density lipoprotein-cholesterol should be used. High density should be changed to high-density.

Responses: Thank you very much for your confirmation of our study. And thank you very much for pointing this issue out. We have revised the text accordingly with green highlight.

“dyslipidemia (elevation of low-density lipoprotein-cholesterol, triglycerides (TGs), and total cholesterol, or reduction of high-density lipoprotein-cholesterol)” (Line 115-117, page 3)